# Position: Make Planning Research Rigorous Again!

**Michael Katz** [1]  **Harsha Kokel** [2]  **Christian Muise** [3]  **Shirin Sohrabi** [1]  **Sarath Sreedharan** [4]

## Abstract

In over sixty years since its inception, the field of planning has made significant contributions to both the theory and practice of building planning software that can solve a never-before-seen planning problem. This was done through established practices of rigorous design and evaluation of planning systems. **It is our position that this rigor should be applied to the current trend of work on planning with large language models.** One way to do so is by correctly incorporating the insights, tools, and data from the automated planning community into the design and evaluation of LLM-based planners. The experience and expertise of the planning community could play a crucial role in accelerating the development of LLM-based planners. This position is particularly important in light of the abundance of recent works that replicate and propagate the same pitfalls that the planning community has encountered and learned from before. We believe that establishing practices that avoid such known pitfalls will contribute greatly to the progress in building LLM-based planners and to planning in general.

## 1. Introduction

With the increased interest in language models came the increased interest in the problems where language models do not perform well. One such problem is planning, which could be informally described as sequential decision-making in the presence of information about the model, i.e., information about the task and the environment. While more efforts are being made to tackle the problems using tools built around large language models (LLMs), it is worth keep-

ing in mind that planning is one of the oldest established subfields of artificial intelligence. The field has taken its roots in the 60's of the last century, with the development of best-first search algorithms (Hart et al., 1968) as part of the famous Shakey project, as well as a formal language to represent planning problems (Fikes & Nilsson, 1971). Since then, the planning community has been developing both the representation languages and the generic solvers that handle problems represented in these languages. Instrumental in the development of these tools was the International Planning Competition (IPC), a series of events that put the state-of-the-art to the test. The IPC had an immense effect on the development of planning systems, but also on the languages used to describe planning tasks, evaluation metrics used, the language features supported, and the development of other tools, such as parsers, grounders, and validators.

Since its inception, the field has made significant contributions to both the theory and practice of building planning software that can solve a never-before-seen planning problem. The main reason it could be achieved is the *rigor with which the research was conducted* on the topic, with methodologies developed over the years through trial and error. This brings us to our primary position: **"Make Planning Research Rigorous Again!"** We believe that this rigor is the cornerstone of thoughtful and reproducible research that can be built upon. Therefore, our belief is that **insights, methodologies, tools, and data from the automated planning community should be correctly incorporated into the design and evaluation of LLM-based planners**.

Lessons learned by the planning community over the decades can help accelerate the development of LLM-based planners. This position is particularly important in light of the abundance of recent works that replicate and perpetuate errors previously made and subsequently addressed by the planning community. At the same time, there is immense promise in LLM-based planning, especially for natural-language interfaces, open-world settings, and tasks where modeling is itself a challenge. Our point is not that such work should stop, but that it should be grounded in rigorous methodology, creating a foundation for future work. Prior work has documented recurring methodological issues in LLM-based planning (Katz et al., 2024). For example, evaluating on scraped public instances creates contamination concerns, and follow-up evaluation on generated in-

---
[1]IBM T. J. Watson Research Center, Yorktown Heights, NY, USA [2]IBM Silicon Valley Lab, San Jose, CA, USA [3]Queen's University, Kingston, Canada [4]Colorado State University, Fort Collins, CO, USA. Correspondence to: Michael Katz <michael.katz1@ibm.com>.

*Proceedings of the 43$^{rd}$ International Conference on Machine Learning*, Seoul, South Korea. PMLR 306, 2026. Copyright 2026 by the author(s).

stances (Katz et al., 2025) provides evidence that benchmark overlap can substantially inflate apparent planning performance. At the same time, some recent work already adopts more planning-aware methodology (Verma et al., 2025; Valmeekam et al., 2025; Huang et al., 2025). Helping the greater AI community to avoid known pitfalls will contribute greatly to the progress in building LLM-based planners and to planning in general.

A position paper, by its very nature, cannot meaningfully summarize all the lessons identified for over half a century. Nor is this paper intended as a systematic survey of LLM-based planning methods; several comprehensive surveys already provide such overviews (Pallagani et al., 2024; Wei et al., 2025; Aghzal et al., 2025; Cao et al., 2025). Our goal in this paper is to provide key pointers and basic vocabulary to help readers find the right resources for their research. We also offer general guidelines for evaluating planners and highlight commonly used community tools useful for developers of LLM-based planners. Finally, we hope this paper serves as a helpful reference for reviewers assessing such work.

## 2. Alternative Views

We outline three positions that oppose the paper's thesis and explain how (and where) they challenge our position.

### 2.1. LLM Planning Is a Different Problem

**Steelman argument.** LLM-based planning is often carried out in open-world, natural-language settings where the state/action space is underspecified, goals shift, the environment may be partially observable, and the transition dynamics are not explicitly provided. Under these conditions, enforcing classical evaluation norms (e.g., strict plan validity, soundness/completeness guarantees) may mis-measure progress or penalize systems that are useful in practice.

**When this view is correct.** This objection is strongest when the task is intrinsically underspecified such that no crisp semantics exists for states/actions/goals, making correctness non-binary and non-auditable.

**Our response.** These tasks fall outside standard planning until formalized. The right scientific move is to identify a well-defined variant of the planning problem, making correctness checkable. This allows to measure the planning progress and identify the correct baselines for comparison.

**Implications.** Relevant planning formalisms should be identified and methods positioned within the existing literature.

### 2.2. Utility Beats Guarantees; Success Is What Matters

**Steelman argument.** Practical planning systems are often *heuristic* and are paired with independent validation or execution feedback loops, yielding a practically reliable overall system. From this perspective, requiring that compared methods provide identical formal guarantees can be seen as overly rigid: what matters is empirical performance (coverage, quality, robustness) under matched resource budgets, with clear reporting of failure rates and costs.

**When this view is correct.** A common justification is that unsound components can be made reliable by pairing them with a validator and repeating until a valid solution is found. This *generate-and-validate* (or rejection sampling) logic is practical when the probability of sampling a valid solution is not too small: either (i) valid solutions are relatively dense in the candidate space, or (ii) the generator places substantial probability mass near valid solutions, typically when solutions resemble previously observed patterns.

**Our response.** In many planning settings valid solutions may be extremely sparse due to combinatorial growth with horizon and strict feasibility constraints, so the expected number of samples required for acceptance can explode, making this wrapper ineffective as a general remedy.

**Implications.** When a method relies on validator-wrapped generation, one should ask whether success is driven by (i) a non-rare solution space under the generator, or (ii) benchmark overlap with training data. The former requires acceptance-rate and budget reporting; the latter requires novelty controls (provenance statements, fresh instance generation, and structure-aware splits) before broad claims about planning can be supported.

### 2.3. Contamination Is Inevitable; We Should Use Harder, More Robust Evaluations Instead

**Steelman argument.** Because modern foundation models are trained on broad public corpora, it may be unrealistic to guarantee that widely used planning benchmarks (and sometimes even their solutions) were unseen during training. Consequently, focusing on "clean" static test sets can be brittle and may mischaracterize progress. A better response is to adopt evaluations that are robust to memorization: fresh instance generation, held-out generation regimes, distribution shifts, and interactive or end-to-end tasks intended to reflect realistic deployment conditions.

**When this view is correct.** This perspective is most compelling when benchmark provenance is uncertain, when solutions are publicly accessible, or when performance improvements plausibly reflect memorization or template recall rather than generalization. In such settings, using generator-backed instances and out-of-distribution regimes can provide stronger evidence than repeatedly scoring on fixed, widely circulated benchmarks.

**Our response.** However, "harder" benchmarks often introduce two failure modes that can undermine scientific

interpretability: (i) *ill-defined or unreliable validation*, where correctness is no longer auditable and is replaced by subjective/model-based judging; and (ii) *capability entanglement*, where a single benchmark score conflates multiple skills (e.g., language parsing, world-model induction, tool use, retrieval, planning, and even formatting), making it unclear what capability is being measured or improved.

**Implications.** Robust-to-contamination evaluation must remain *well-defined* and *diagnostic*: state the target formalism and solution concept, use auditable validation whenever available, and avoid opaque "kitchen-sink" benchmarks by reporting stage-wise results or controlled variants that isolate capabilities. Always report full budgets (time/memory, LLM/tool calls, retries/repair) and use generator-backed, structure-aware splits for novelty and scaling.

**Connection to our position.** We agree that evaluations must be robust to contamination, but argue that "harder" should not mean less well-defined or less diagnostic. The planning community already provides mature protocols for this: semantics-preserving benchmark specifications, independent validation when available, generator-backed instance creation with difficulty scaling, and evaluation/reporting practices that separate capability sources and account for budgets. Our position is that LLM-based planning work should adopt these established protocols rather than replacing them with opaque, mixed-capability benchmarks that obscure what is being measured.

## 3. Fundamental Planning Terminology

In this section, we provide a brief overview of different flavors, formalism, and representations of planning problems commonly used by the AI Planning community.

We start from the most basic formalism, commonly referred to as classical planning. A classical planning problem includes the initial state of the world, desired goals, and a set of possible actions. The objective of a planner is to synthesize a plan that is guaranteed to generate a state which contains the desired goals. Formally, the classical planning model is defined as a tuple $\mathcal{S} = \langle S, s_0, S_G, A, f, c \rangle$, where

- $S$ is a finite and discrete state space
- $s_0 \in S$ is a known initial state
- $S_G \subseteq S$ is a set of goal states
- $A$ is a set of actions; $A(s) \subseteq A$ applicable in each $s \in S$
- $f$ is a deterministic transition function, where $s' = f(a, s)$ for $a \in A(s)$
- $c$ is a non-negative action costs, $c(a, s)$.

The solution to a classical planning problem is a sequence of applicable actions that maps $s_0$ into $S_G$.

In the classical setting, the initial state is known, actions are deterministic, the system is fully observable, actions are instantaneous, goals are specified as final states, and the planning horizon is finite. Relaxing these assumptions introduces multiple variants of planning, including:

- Conformant planning (Smith & Weld, 1998; Bonet & Geffner, 2000) - where initial state is uncertain.
- Probabilistic planning (Yoon et al., 2008; Domshlak & Hoffmann, 2007; Littman, 1997; Sanner, 2010) - where the action dynamics is probabilistic and state spaces do not have to be discrete or finite.
- Non-deterministic planning (Kuter et al., 2008), and fully observable non-deterministic planning (FOND) (Mattmüller et al., 2010; Muise et al., 2012; 2014) - where the action dynamic is non-deterministic.
- Temporal Planning (Fox & Long, 2003; Gerevini et al., 2010; Long & Fox, 2003) - where actions have durations and temporal constraints.
- Numeric Planning (Helmert, 2002; Piacentini et al., 2015; Srivastava et al., 2011) - where actions can affect numeric state variables.
- Hierarchical Task Network (HTN) planning (Erol et al., 1994; Nau et al., 2003) - where the initial state and the goal are defined as a task network (a set of tasks and constraints), with recursive decomposition of high-level tasks into lower-level sub-tasks.
- Planning with soft or hard constraints: preferences (Baier & McIlraith, 2008; Sohrabi et al., 2009), temporally extended goals (Bacchus & Kabanza, 1998; Baier & McIlraith, 2006) - where the goal is to additionally optimize for the soft and hard constraints, as well as partial satisfaction planning, such as net-benefit (van den Briel et al., 2004; Keyder & Geffner, 2009; Aghighi & Jonsson, 2014), where the difference between the utility and the cost of achieving the goal is optimized, and oversubscription planning (Smith, 2004; Domshlak & Mirkis, 2015; Katz & Mirkis, 2016) - where a solution is a plan that achieves the best possible utility given resource constraints.

Several of these flavors of planning can be compiled into classical planning for easier computation: conformant planning (Palacios & Geffner, 2009), a fragment of HTN planning (Alford et al., 2009), and compiling away soft goals/preferences (Keyder & Geffner, 2009).

While most of these models can be captured by the well-known (PO)MDP formalism, it is more efficient to capture in a compact way (e.g., with a factored representation), with a formal planning language. Multiple such formal languages for planning problems have been introduced. Starting with ground representations, the most popular are the simplest propositional language STRIPS (Fikes & Nilsson, 1971),

f-STRIPS (Geffner, 2000), all the way to multi-valued variables based languages SAS/SAS+/FDR (Bäckström & Klein, 1991; Bäckström & Nebel, 1995; Helmert, 2009).

This variety, however, is a mixed blessing, as various existing solvers that were being developed supported one language or the other, making it more difficult to compare their performance. Consequently, with the birth of International Planning Competitions in 1998, a unified language for planning problems representation was born, named Planning Domain Definition Language (PDDL) (McDermott et al., 1998). This language has become the de facto standard and most commonly used representation language for planning. The planning knowledge is separated into two parts, the PDDL domain and the PDDL problem. PDDL domain contains knowledge of the constants, types, predicates, actions, their preconditions, and effects. PDDL problem consists of knowledge of the objects, the initial state, and the goal state. PDDL has many extensions (Haslum et al., 2019). Furthermore, RDDL (Sanner, 2010) and PPDDL represent probabilistic planning tasks (Younes et al., 2005).

Compact and accurate representation is challenging to derive. As a result, learning planning models has become a growing area of interest over the past decade (Yang et al., 2007; Hogg et al., 2010; 2008; 2009). Learning the PDDL representation, has gained special attention in the era of LLM. The existing work can be partitioned into learning the domain model (Guan et al., 2023; Oswald et al., 2024; Gestrin et al., 2024; Tantakoun et al., 2025) and learning the problem instance representation, assuming the domain model exists (Liu et al., 2023; Zuo et al., 2025).

**References to Some Tutorials:** Readers can use the following tutorials as starting points to read more about many of the topics discussed above: on classical planning, on the Python library Unified Planning that supports modeling and programmatically invoking planners, on RDDL, on methods to generate multiple plans, and on learning planning models.

For additional material, we refer the curious reader to the textbooks (Ghallab et al., 2004; 2016; Mausam & Kolobov, 2012; Haslum et al., 2019).

**Common Pitfall 3.1** While planning problems can have infinite state spaces, they are well-defined mathematically. Problems with fuzzy and ill-defined state and action spaces are generally not considered by the planning community.

**Action:** Identify a well-defined variant of the problem, allowing also to define what constitutes a solution.

**Common Pitfall 3.2.** A problem being representable as a planning problem does not imply that planning tools are a good fit for it. For example, many reasoning problems, like the canonical NP-complete problem of boolean satisfiability (SAT) (Cook, 1971) and puzzles like Sudoku (Simonis,

2005; Lynce & Ouaknine, 2006), can be represented as planning problems. However, SAT and CSP solvers would be better suited to serve as baselines for these problems.

**Action:** Check if your problem has sequential nature where the order of decisions matters. If not, consider whether SAT/CSP solvers might be more appropriate baselines.

## 4. Problems, Algorithms, and Complexity

Even just for classical planning, there is a variety of decision and search problems one can think of. Two decision problems are the most popular, *plan existence* and *bounded plan existence*, a variant that asks whether the plan exists of quality under a given bound. There is a larger variety of search problems, including *cost-optimal planning*, asking to find a plan that minimizes summed action cost, or, in the case of unit-cost actions, minimizes plan length. Other search problems include *satisficing planning*, asking to find any plan, while cheaper plans are better; *agile planning*, where the only thing that matters is how quickly the plan is found; *top-k planning*, asking to find k plans such that no cheaper plans exist; *top-quality planning*, asking to find all plans up to a certain cost; and *diverse planning*, aiming at obtaining diverse set of plans, considering plan quality as well. A planner that is guaranteed to only return a valid plan is a *sound* planner, and a planner that is guaranteed to return a solution if one exists is a *complete* planner.

While all the problems described previously are PSPACE-hard (Bylander, 1994) in general, when represented in a planning language such as PDDL, for some domains, the complexity can significantly vary from one problem to the other. One prominent example is the BlocksWorld domain, where cost-optimal planning is NP-complete, while agile or satisficing planning are in P. To see that, one can think of a simple two-stage policy that can solve any BlocksWorld instance - simply unstack all blocks and put them on the table in the first stage and incrementally build the requested goal state in the second stage. Such policies are called *generalized policies* and are dealt with in the generalized planning subfield. Generalized policies solve planning problems without any search, and while finding these policies may be useful, possibly with language models (Silver et al., 2024), one should be careful generalizing the evidence obtained on these problems to the entire field of planning.

When looking at individual domains or problems, one must ask oneself, how difficult is it to solve? In cases when the semantics of the domain are known, some computational complexity results exist for individual domains, e.g., (Helmert, 2006a; Culberson, 1997; Dor & Zwick, 1999; Hearn & Demaine, 2005). In other cases, a large body of research investigated the computational complexity of different fragments of planning, mostly based on the structure

of the ground planning task, such as causal graph structure, variables domains size and structure, reversibility/invertability of actions, action preconditions size, and the mixture of those, e.g., (Bäckström & Klein, 1991; Bäckström & Nebel, 1995; Bylander, 1994; Jonsson & Bäckström, 1998; Helmert, 2003; Katz & Domshlak, 2008; Domshlak et al., 2015). The rationale behind the investigation, in addition to understanding where the boundary between easy and hard problems lies, was in using the poly-time fragments for automatically deriving poly-time computable distance to goal approximations, also called *heuristic functions* (Hoffmann & Nebel, 2001; Edelkamp, 2001; Katz & Domshlak, 2010). Other difficulty approximations include various notions of *width*, meant to capture how hard it is to solve a problem, with the goal of exploiting the property algorithmically (Chen & Giménez, 2007; Lipovetzky & Geffner, 2012; Dold & Helmert, 2024; Mao et al., 2023). While a full formal characterization is not always available or necessary for every practical benchmark, understanding the complexity and guarantees of the target problem is often important for selecting fair baselines and interpreting empirical results.

To solve these problems, a variety of methods were introduced, starting with the famous STRIPS algorithm (Fikes & Nilsson, 1971), a total ordering planning backward from the goal, keeping a stack of subgoals, iteratively resolving a top subgoal. The algorithm is sound but incomplete for satisficing planning, but a powerful planner from the 70s nonetheless. The 80s gave rise to methods based on Partial Order Planning, a search in the space of partial plans, a sound and complete approach to satisficing planning. The 90s were the Golden Age of classical planning, introducing four different approaches to planning: GraphPlan, SAT-plan, heuristic search planning, and model checking planning. GraphPlan (Blum & Furst, 1995) constructs a graph containing all possible parallel plans up to certain length in a forward manner and then extracts a plan by searching the graph backward from the goal. This is a sound approach to satisficing planning, with the completeness obtained by increasing the length bound. The other three approaches were not just both sound and complete. Importantly, these approaches could produce optimal solutions. SAT-plan (Kautz & Selman, 1996) exploited SAT solvers, transforming the planning task into a boolean formula. It optimized for plan length by increasing its bound. Model Checking Planning (Cimatti et al., 1997) constructed a layered graph, compactly representing multiple states in a layer with compact data structures for representing boolean functions. Then, a backward search is performed from the goal to find a plan. Finally, the most popular approach to date is heuristic search planning, a forward search in the problem state space, with automatically extracted from the problem heuristic function. Over the years, a large variety of search algorithms and heuristic functions were introduced, some

guaranteeing the obtained solutions to be optimal. One such famous algorithm is A$^*$, a best-first search algorithm that expands the search nodes in the order of their $f = g + h$ values, where $g$ is the cost of getting to the node from the initial state and $h$ is the heuristic function value for the node. When the heuristic function is *admissible* (guaranteed not to over-estimate the true cost of getting to the goal), the algorithm is guaranteed to produce cost-optimal plans. One additional property of the algorithm is that it is optimally efficient (Dechter & Pearl, 1985), meaning **there cannot be any algorithm in its class that can do better than A$^*$**. When going beyond classical planning problems, popular approaches include the aforementioned transformations to classical planning and use of classical planners, as well as the use of more complex heuristic search algorithm families, such as MCTS/UCT or And/Or best-first search.

Another important aspect is the complexity of solution validation. In classical planning, solutions (aka plans) are applicable to the initial state sequences of actions that result in a goal state. Validating whether a sequence of actions is a plan is poly-time, but in order to validate that a plan is cost-optimal, one needs to prove that there is no plan of smaller cost, which is PSPACE-complete. In HTN planning, any solution validation is NP-hard. For non-deterministic planning, solutions are typically represented as policies mapping states to actions or compactly as controllers, and validating their complexity is poly-time in the solution size.

**Common Pitfall 4.1.** Historically, proposed planning algorithms were at least sound, albeit sometimes incomplete, meaning when they produce a solution, this solution is guaranteed to be correct. Naturally, an implementation may include bugs, and therefore it is a good practice to validate the solutions produced. Unsound algorithms do not have the guarantee to output only correct solutions, and therefore must be followed by a validation.

**Action:** Add a sound validator to your pipeline to verify that solutions produced by unsound algorithms are correct.

**Common Pitfall 4.2.** It is a common practice to compare planners that target the same computational problems. In most cases, a comparison between planners built for different computational problems—thus providing different guarantees on the produced plans—is less meaningful.

**Action:** Select baselines that provide the same guarantees.

**Common Pitfall 4.3.** It is important to know the properties of the algorithms used and of the resulting solutions. One common error is to use A$^*$ search with a heuristic function that has no admissibility guarantees as a planner. In such cases, there is no guarantee for a produced plan to be cost-optimal, and validating cost-optimality can be extremely resource consuming. Further, while A$^*$ is optimally efficient for cost-optimal planning, it is extremely inefficient

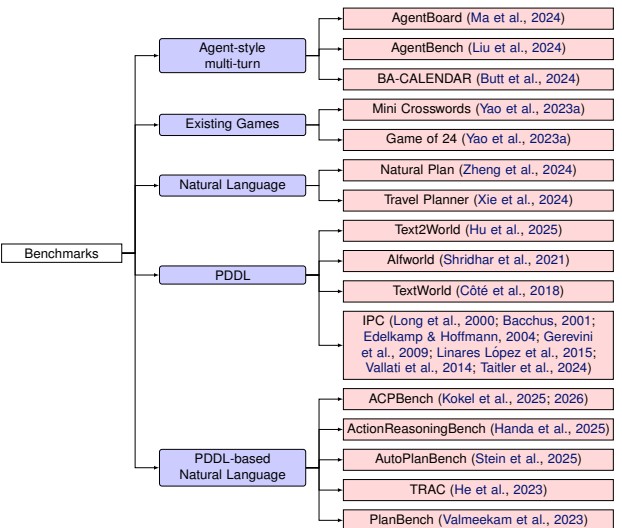

*Figure 1.* Benchmarks for LLM-based Planning evaluations

for satisficing planning, when no optimality guarantees are needed. This is due to the fact that it spends most of its efforts attempting to prove that there are no better plans.

**Action:** Choose the appropriate search algorithm for your problem type. Examples: A* with admissible heuristic for optimal planning, greedy best-first or other algorithms for satisficing planning.

**Common Pitfall 4.4.** While regression from goal (Reiter, 2001) is a valuable tool in planning, one must know that the process creates so-called *spurious* states, which can be invalid or simply unreachable from the initial state. In both cases, one should take great care with such states.

**Action:** Beware the limits of regression-based approaches.

## 5. The Data

The majority of datasets used for evaluation of LLM-based planners fall into one of the following categories: (a) existing games repurposed for multi-step planning/search problems, (b) natural language planning problems generated specifically for LLM evaluations, (c) existing PDDL domains, and (d) natural language datasets generated from PDDL domains. Figure 1 provides an overview of various benchmarks. Current LLM-planning work spans tasks that vary substantially in how directly they align with standard planning formalisms, and this affects which claims, baselines, and evaluation protocols are appropriate. In this section, we offer insights into the PDDL benchmarks creation and highlight a few common pitfalls and misconceptions in benchmark selection for evaluations. We propose a few key considerations to guide a robust and meaningful assessments of planning systems.

**Common Pitfall 5.1.** International Planning Competitions (IPC) played a pivotal role in the development of planning systems. The intention behind IPC was to test the planners on unknown, previously unseen benchmarks. The participants therefore submit their planners, which are run by the IPC organizers on a collection of planning problems, using the same hardware, under the same time and memory restrictions. IPC organizers therefore needed to introduce new PDDL domains every time the competition was run. These domains were meant to capture some abstraction of a real life problem, but often also were crafted to double down on known limitations of popular approaches. One example is the *Gripper* domain, an easy problem which introduces many *symmetries* into the state space, making it challenging for pure forward (heuristic) search-based methods. This pushed the research towards investigating search pruning techniques (Fox & Long, 1999; Pochter et al., 2011; Chen & Yao, 2009). Another aspect to consider is the problem instances created for each domain. These problem instances should be challenging enough for the current state-of-the-art, but also not too challenging, so a meaningful comparison would be possible. Consequently, a collection of problem instances of increasing difficulty was created. Initially, domains varied significantly in the number of instances created, the later editions were more uniform, for more meaningful aggregation. To automate the process, a tool to automatically scale the instances was developed (Torralba et al., 2021), allowing us to create a uniform size set of instances of existing domains that are challenging for the current state-of-the-art in cost-optimal, satisficing, or agile planning. **Instances that are challenging for cost-optimal planning might be very easy for satisficing or agile planning, where no optimality is required.**

**Action:** Match benchmark difficulty to your problem type. Use appropriately challenging instances for your specific planning setting (optimal/satisficing/agile).

**Common Pitfall 5.2.** A critical concern of evaluating models on publicly available planning domains and problem instances, such as BlocksWorld, Logistics, and Grid-based IPC benchmarks, is that the solutions are accessible through planning resources (c.f. Muise (2016)). This issue is most aggravated when the benchmark is generated by scraping the data from the internet. For instance, the "Game of 24" and the "Mini crosswords" datasets (Yao et al., 2023a) were generated by scraping `https://4nums.com` and `https://www.goobix.com/crosswords/0505/`, respectively. This raises the risk that answers may be included in the training data, leading to potential memorization and artificially inflated performance estimates. Indeed, the contamination study by Hu et al. (2025) shows that frontier models have memorized the PDDL. This underscores the need for careful attention to data provenance when selecting benchmarks. **Ideally,**

**evaluation should be conducted on novel domains and problem instances that are guaranteed to be unseen during training.** Similar concerns were addressed in planning by introduction of "mystery domains", where predicates and object names were replaced with random words. The use of semantically uninformative or misleading names may introduce ambiguity for language models, potentially impairing their performance. Consequently, such domains may not be suitable for reliably evaluating the performance of language models.

**Action:** Build generators that generate random instances for evaluations. Such generators are readily available for over 60 PDDL domains.

**Common Pitfall 5.3.** Another challenge in developing planning benchmarks is the lack of reliable ground truth and difficulty of verifying the correctness of the answers. In some domains, especially those involving open-ended reasoning, the correctness of a solution may not be binary. To overcome that challenge, certain datasets use LLMs for evaluation. For example, Butt et al. (2024) and Handa et al. (2025) use LLM as a judge for evaluating the generative tasks. While this has become acceptable approach for evaluation for some Question Answering tasks, it is unreliable for planning problems. Another source of uncertainty in the ground truth also arises from the fact that some datasets are generated by scraping the internet and individual examples and their ground truth are rarely verified. The unreliability of the data is most prolonged when they are generated using LLMs. For example: In an earlier version of AutoPlanBench (Stein et al., 2025), '(get-pop ?x)' action in the Movie domain is translated to "get population of an object" instead of getting a soda.

**Action:** Use deterministic validators instead of LLM-based evaluation. Verify ground truth for scraped datasets. Avoid using LLM-generated data without careful verification.

**Common Pitfall 5.4.** Greater care is required to split the data as equivalence and hardness is not immediately apparent from the questions. It is standard practice to *randomly* split generated data into a train-set and test-set for evaluating the performance of a model. While it may work in most cases, for data generated using PDDL-generators, a random split might not work. The PDDL-generator doesn't ensure that the generated problem instances are all unique. The generator might produce multiple problem instances that are structurally or functionally equivalent (i.e., isomorphic). Meaning that two instances might differ only in superficial aspects such as object naming or variable ordering, while preserving the same underlying structure and hence share the same solution strategy. Hence, selecting syntactically unique problem instances is not sufficient to evaluate the underlying reasoning ability of the model. To overcome this issue, some prior works partition the test and train split

based on the number of objects or the plan-length. For example, train set might include problems with 3–7 objects where as test set might include problems with 7–20 objects. This approach ensures that the test instances are structurally different and systematically more complex then the train instances. However, caution is warranted when using such partitioning strategies, as they can give a misleading impression of "correctness". For instance, consider a BlocksWorld domain where test instances are said to include 7–20 blocks. If, in practice, the plan length is clipped at 10 steps, and assuming a 4-operator BlocksWorld domain, the planner can only move a maximum of 5 blocks. In such cases, despite the apparent increase in object count, the solutions for the test problems are structurally similar to those in the training set, undermining the intended evaluation (Saha et al., 2025).

**Action:** Use structure-aware splits (e.g., by plan length or number of objects). Verify that test instances are actually more complex, not just syntactically different. Check that plan length limits don't artificially cap complexity.

## 6. Tools From The Planning Community

Recent years have seen a surge in development of tools for working with planning problems. From off-the-shelf planners to online services to debugging software, there is a whole host of software and libraries to researchers' disposal. Here, we detail some of what is now available.

For the manual specification of PDDL, an online editor is available for quick prototyping and solving of various planning formalisms, and an extended interface is available through the VSCode plugin for PDDL. Both of these services contain an integration with a suite of planners hosted in the cloud for free use (with limited resources). The service is open for programmatic access for anyone wishing to solve simple problems. The service is also available as an open source project for those who wish to host a version with different resources or planners (Ding et al., 2023).

For running planners and planning software directly, the planutils offers turn-key access to dozens of existing software, all pre-compiled (Muise et al., 2022). This allows researchers to forgo the often fraught process of getting planners compiled and running. Not only does this include many state-of-the-art planning systems for various formalisms, but it also includes general planning software, such as model debugging (Käser et al., 2022) and plan validation (Howey & Long, 2003; Haslum, 2016). While planutils is a convenient way to access many tools, it is worth mentioning some of the more popular tools directly. First and foremost, is the most popular planning system Fast Downward (Helmert, 2006b), that implements many search algorithms, heuristics, pruning techniques, as well as methods for storing and accessing search queues. Many state-of-the-art planners are

built on top of Fast Downward and find their way into the core code. As such, invoking Fast Downward with different parameters will result in planners for different formalisms, such as cost-optimal, satisficing, agile, etc. Another popular planning system is Pyperplan (Alkhazraji et al., 2020). Its popularity, however, stems from the ease of use rather than its power. It is important to know that Pyperplan was created for educational purposes and should not be used for experimental comparison. One category of particularly useful planning formalisms focuses on finding multiple solutions to planning problems (Riabov et al., 2014; Katz et al., 2018; 2020; Katz & Sohrabi, 2020; Katz et al., 2022; Speck et al., 2020; Lee et al., 2023; Katz & Lee, 2023a;b). Planners for these formalisms are Forbid Iterative, K*, and SymK. All three can be conveniently invoked from Python. The former two are offered as PyPi packages. The latter is accessible via Unified Planning library (Micheli et al., 2025).

PDDL problems generators offer customizable software to generate problem instances for over 50 unique planning benchmark domains (Seipp et al., 2022). PDDL parsers and grounders allow one to parse, modify, and generate PDDL representations, as well as progress and regress through state spaces. Among the most commonly used are the pddl Python library , the *Tarski* parser (Francés et al., 2018), which can also be used via a lightweight wrapper, as well as CPDDL planning library, which implements a variety of tools for task manipulation. These libraries are absolutely invaluable for creating PDDL content programmatically.

**Common Pitfall 6.1.** It is a good practice to mention which configuration is used, and it is absolutely necessary to mention what search problem is solved. For instance, saying that Fast Downward was used does not provide sufficient information about the planner, it could be a cost-optimal configuration, a satisficing or agile one. It could also be a configuration that combines components in a way that should be justified (e.g., A* search with inadmissible heuristic, refer to Common Pitfall 4.3 for additional discussion).

**Action:** Report full planner configuration details including search algorithm, heuristic, and what problem type it targets (optimal/satisficing/agile), if not clear for a planner.

**Common Pitfall 6.2.** Packages that allow to progress from one state to another via action application are based on the process of *grounding* a planning task. As the tools are oriented towards creating planning tasks for an efficient search, they often get rid of irrelevant to the search process information, such as so-called *static* information, that does not change from one state to another, like game maps or object types. While symbolic planners do not need the information once the task is grounded, LLM-based planners might find this information crucial. Being aware of how the tools work may help preserve the needed information.

## 7. On Evaluating Planners

In this section, we put everything together to describe what typically constitute a rigorous scientific evaluation of a planning method, or simply a *planner*. When proposing a new planner, it is important to clearly articulate the assumptions made in terms of the planning problems being addressed. As we discussed in Section 3, planning problems vary widely, each consider different assumptions. Clearly stating these assumptions allows the authors to accurately position their contribution within the relevant literature, select appropriate baselines for evaluation, and most importantly avoid claims that are too general and are not supported by the scope of the work. Our overview in Section 3 is by no means complete, but can serve as a useful starting point for researchers to better understand the extensive AI Planning literature and to properly situate their contributions.

**Common Pitfall 7.1.** It is critically important to have an understanding of the computational complexity of the problem at hand. For any newly proposed planner, it is standard practice to provide the complexity analysis of the planner, along with formal properties such as soundness, completeness, and, where applicable, optimality (Katz et al., 2024). When a full formal complexity characterization is unavailable, one should still clarify the guarantees being targeted and the basis on which baselines are considered fair. These considerations are particularly important when selecting baselines. In general, unsound methods should be avoided, or, at the very least, not compared directly to sound approaches, as they do not provide the same guarantees. For newly proposed domains/datasets, if no validator exists, it is important to provide a sound validator.

**Action:** Identify the complexity and fundamental properties of the planner used, and align the baselines appropriately.

**Common Pitfall 7.2.** The most common evaluation metric is the overall aggregated coverage, where the coverage per instance is 1 if solved, otherwise 0. This is similar to the common success rate metric. While in principle this metric works for many computational problems, both optimal and non-optimal settings, the metric is more suitable for optimal settings, when all found plans should be of the same quality. For satisficing and agile settings a different metric was proposed, known as the *IPC score*. This metric scores the method relatively to other competitors performance per instance. For satisficing setting, each instance gets the value $\frac{c^*}{c}$ where $c^*$ is the best among the known plan costs, and $c$ is the cost of the plan obtained by the method. For agile planning, the time to get the solution is used instead of the plan cost.

**Action:** In cases when a search component is proposed, report a measure of search effort, such as the number of expanded nodes for optimal search algorithms and generated

nodes for satisficing search. Evaluate on a large variety of domains to reduce the bias to particular domains. Compare methods under the same hardware and resource restrictions.

**Common Pitfall 7.3.** When proposing planners that do not have soundness guarantees, as it common in LLM-based planning literature (Wei et al., 2022; Yao et al., 2023b;a; Sel et al., 2024; Gandhi et al., 2024), the planner must be paired with a *sound* validator, which can make the overall algorithm technically sound. When validation logic is embedded in task-specific code rather than separated as an independent planning-style validator, it may fail to check whether actions used are from the set of allowed actions, leading to possibly falsely validated outcomes. This pattern has been observed in some LLM-based planning work (e.g., Yao et al., 2023a).

**Action:** Separate the validation from the solution to reduce the possibility of error.

**Common Pitfall 7.4.** In the cost-optimal track of the IPC, when the planners must provide optimal solutions, participants that return non-optimal solutions are disqualified. Recall that it is hard to validate that a solution is cost-optimal, and therefore we are meant to *trust* the cost-optimal planner. It does not matter how often the solution produced *happens to be* cost-optimal, if no such guarantee exists, and therefore it is meaningless to measure (Lehnert et al., 2024).

**Common Pitfall 7.5.** The search effort comparison is meaningful only when the same search algorithm is used, evaluating the relative power of the proposed heuristic function or pruning technique compared to existing ones. Further, this should be a reasonable algorithm for the problem at hand. For example, $A^*$ is not a reasonable choice of an algorithm for non-optimal planning, as it puts more effort in proving optimality than finding a plan. If, in addition, the proposed heuristic function is not admissible, then this additional effort is essentially *wasted*, as optimality of the found solution cannot be guaranteed. Comparing the search efforts across algorithms might be tricky, even if they provide the same plan quality guarantees. If the plan quality guarantees are different, the comparison is meaningless, as they solve different problems, sometimes of different complexity (recall the discussion of BlocksWorld in Section 4). Further, some algorithms, like $A^*$ are optimally efficient, see Section 4. Thus, one might want to avoid claims of *"Better Planning"* based on such observations (Lehnert et al., 2024).

**Common Pitfall 7.6.** The choice of datasets for evaluation is extremely important. First, the dataset must be relevant to the problem at hand and suitable for evaluating the proposed approach as well as the baselines. Moreover, the performance should be evaluated across a variety of domains, showing how the proposed method scales with increasing problem instance difficulty. Many existing datasets, such as those in (Yao et al., 2023a), consists of instances of similar

size and complexity (e.g., all instances of the 24 game are of the same size), which limits the generality of the evaluation. Second, care must be taken to decouple the proposed method from the dataset to ensure unbiased evaluation. **Unfortunately, it has become common practice to introduce both a new dataset and a new method in the same paper, potentially leading to evaluation bias.** IPC benchmarks were established to help mitigate this issue by providing standardized evaluation domains. Please refer to Section 5 for further discussion on pitfalls related to dataset selection.

**Common Pitfall 7.7.** A key aspect of high-quality scientific paper is the transparency of the experimental setup. This includes clearly specifying the computation resources used, time limits, the number of LLM calls, and all the tools involved in both the proposed approach or baseline approaches. A frequent issue is the insufficient specification of the planner (see Common Pitfall 6.1), or compute resources. Such oversights make reproducibility hard.

# 8. Conclusions

With increased focus on solving planning problems using LLM-based tools, it is now more important than ever to approach this problem in a systematic and rigorous way. This would not only help us take an accurate stock of the state-of-the-art, but also prevent us from fooling ourselves into thinking we have made more progress than we actually have. To that end, we propose to adopt insights, methodologies, tools, and data from the automated planning community and incorporate them in the design and evaluation of our AI systems. Our highest-priority recommendations are to

1. clearly specify the planning problem being addressed,
2. use independently checkable evaluation whenever possible,
3. compare against strong relevant baselines, and
4. evaluate on diverse benchmarks with difficulty scaling and novelty controls.

Automated planning community provides us with a useful framework to categorize and analyze the various flavors of planning, including identifying their computational complexities. The field has not only developed an extensive set of benchmarks that could be directly adopted by researchers working on LLM-based planners, but also developed helpful tools and established useful experimental protocols. This paper also details basic terminology and pointers that could serve as a starting point for researchers to learn more about existing work and the state of the art. In each section, we also laid out advice for researchers less familiar with the field, as well as common pitfalls to avoid. Ultimately, we hope this paper serves as a handbook for conducting rigorous, planning-based research for researchers and reviewers alike.

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
