# OpenReview forum: "Position: Make Planning Research Rigorous Again!"
_ICML.cc/2026/Position_Paper_Track — ICML 2026 Position Paper Track regular_

### Official Review · Reviewer_JoQZ · 2026-03-10

**Significance:** 3
**Argument Clarity:** 3
**Rating:** 5
**Confidence:** 4

**Questions:**

C1. Regarding Common Pitfall 7.1: while understanding complexity is the norm in your field, it may not be universally critical; many practical problems elude formal complexity characterisations while clearly being computationally challenging. Can you support this with arguments and not the norms of your field? This also overlaps with 7.6; same point is made at the end in each.

C2. Regarding Common Pitfall 7.6: I do not necessarily think that introducing a new dataset and a new method is an issue, *if* the claims of the paper extend to those benchmarked problems only. After all, the authors themselves argue that problems in IPC were designed explicitly to challenge limitations of existing planners. Is this not also an instance of designing planners that perform well on those specific benchmarks, rather than in general?

C3. Typos and writing suggestions
- L060: "Steelman" -> "Steelman argument"? Some readers may be unfamiliar with the terminology.
- L130 "costs" -> "cost"
- L116 RHS: "and the and"
- L161: break up last bulletpoint, hard to parse
- L126: Worth giving citations to POMDP/MDPs, ditto for MCTS/UCT (L238)
- L188: bound on what?
- L190 RHS: the chronology paragraph is much too long and should be broken down
- L240: complexity was discussed earlier, why not move with that content?
- L276 RHS: citation bracketing
- L409 RHS: "or or"

**Alternative Views Section:**

Yes

**Compliance With Llm Reviewing Policy A Conservative:**

Affirmed.

**Discussion Potential:**

3

**Final Justification:**

The rebuttal has addressed my key concerns. The resulting modifications promised by the authors, while non-trivial, are addressable in the camerea-ready revision cycle. While there are some weaknesses, on the whole I believe that the community would benefit from this paper.

**Paper Summary:**

This paper argues for the position that planning (a field in AI with a long history) should be made rigourous again. This position is taken in light of recent works addressing planning problems with LLM-based approaches. The authors argue that these studies fall into a series of pitfalls that have been experienced by the planning community in the past. It is proposed that by adopting the formalisms, benchmarks, and evaluation procedures of the planning community, more reliable and rapid progress can be made in this emerging field. The paper introduces fundamental planning terminology, discusses problems and benchmarks, software tools, and evaluation methodology; going into details on common pitfalls and challenges along the way.

**Position:**

Yes

**Position In Title:**

Yes

**Related Work:**

2

**Strengths And Weaknesses:**

## Strengths

S1. The paper is very well-structured and well-written.

S2. The paper has very good scholarly quality and, in my view, supports its arguments in a fairly compelling fashion. It also gives in-depth consideration to alternative views and explores supporting arguments for those.

## Weaknesses

Some aspects I judge as being weaknesses of the work are below. However, on the whole, I consider that the ICML community would benefit from this position paper.

W1. The paper currently reads as a hybrid between a position paper and an introductory tutorial to the key concepts of planning and some important resources in the community.
- As such, in my opinion, it waters down the strength of the position taken by the authors, as significant space needs to be dedicated to the "tutorial" parts; and there is also insufficient room to introduce the planning concepts in meaningful depth.
- While I understand why this choice was made, the tutorial part would perhaps be more helpful as a separate, more expansive resource (e.g., monograph, series of blog posts).
- As somebody who works with Markov Decision Processes and has some familiarity with (classical) planning, I doubt that the paper is accessible for newcomers to the field, given the density of the jargon.

W2. With a few exceptions, the work tends to not cite papers in the planning-with-LLM literature (it is sufficient to do a cursory scan of the bibliography to verify this). I think this leads to two issues:
- The "pitfalls" raised by the authors generally lack citations to example papers falling prey to said pitfalls. The assertion that LLM-based planning research should be supported with such citations.
- If the goal is to get the LLM-based planning community to notice the classical planning work, citing them is a good place to begin.

W3. Lastly, I believe the paper fails to acknowledge the advantages of LLM-based planning, as the field is painted in an exclusively negative light.
- Why have such works emerged in the first place? Is it perhaps due to the limitations of current planning techniques e.g. inability to deal with open-endedness, difficulty of specifying the tasks in formal language, poor accessibility of terminology and tools in the field?
- If LLM-based approaches are not the answer, how might these limitations be addressed by the field otherwise?
- Labelling this as "not planning" (Section 2.1; Pitfall 3.1) is, in my opinion, insufficient.

**Support:**

2

---

> ### Author Rebuttal · Authors · 2026-03-31
>
> We thank the reviewer for the constructive feedback.
>
> **On Weaknesses:**
>
> We believe that there is immense potential in using language models for planning. The ideas that the scientific community has come up with so far are diverse, creative, and full of potential. The main point we want to make in this position paper is that these ideas, implemented and evaluated rigorously, will yield stronger outcomes and provide reliable building blocks for the scientific community to use and build on. We will try to adapt the paper's rhetoric to better reflect this position.
>
> Additionally, it is worth noting the many LLM planning papers that already do a great job in this regard. Regrettably, the way our paper was structured did not leave much space to highlight specific work that avoids the pitfalls mentioned. We are considering adding a paragraph that mentions some of these works. Here is a small subset of such examples.
> [1] Pulkit Verma, Ngoc La, Anthony Favier, Swaroop Mishra, Julie Shah, PDDL-INSTRUCT: Enhancing Symbolic Planning Capabilities in LLMs through Logical Chain-of-Thought Instruction Tuning, LM4Plan@ICAPS 2025
> [2] Karthik Valmeekam, Kaya Stechly, Atharva Gundawar, Subbarao Kambhampati, A Systematic Evaluation of the Planning and Scheduling Abilities of the Reasoning Model O1. Transactions on Machine Learning Research, 2025
> [3] Sukai Huang, Trevor Cohn, Nir Lipovetzky, Chasing Progress, Not Perfection: Revisiting Strategies for End-to-End LLM Plan Generation, The International Conference on Automated Planning and Scheduling (ICAPS), 2025
>
> **C1:**
> We agree that identifying the complexity of every potential benchmark may not be necessary. However, identifying the complexity of the benchmarks you are working with could help you glean deeper insights from the evaluations you perform on those benchmarks.
>
> A simple example is the Blocksworld domain, which is polynomial for plan generation but NP-hard for optimal plan generation. If you compare a non-optimal planner to an optimal one on blocksworld, you put an optimal planner at a disadvantage. If you are comparing the plan quality, you might even see the non-optimal planner returning optimal solutions for the tasks you tested on, but that does not mean you can *rely on it to produce optimal solutions*, while the optimal planner will give you such guarantees, but will spend considerably more effort doing so.
>
> Indeed, 7.1 and 7.6 arrive at the same point from different angles. This is an important point, disregarding which, as was for example done by Yao et al, NeurIPS 2023, leads to reported performance that was possibly skewed due to memorization concerns (the dataset was scraped from the internet). A recent work [1] shows that simply moving to different, generated instances reduces the accuracy of the Tree of Thoughts approach from around 80% to around 20%.
>
> [1] Katz et al, Seemingly Simple Planning Problems are Computationally Challenging: The Countdown Game, LM4Plan@ICAPS 2025
>
> **C2:**
> The IPC domains were designed separately from the planners' designs to challenge existing planners, not planners submitted to that IPC.
> Having said that, we certainly agree that if a paper introduces an approach that tackles a specific issue, and there are no existing benchmarks that allow for testing that issue, one can and should introduce such benchmarks. However, this case is uncommon. The more common case is that multiple fitting benchmarks exist for the authors to test their approaches, and showing the method's performance on a variety of already existing benchmarks strengthens the author's case.

---

> > ### Author Rebuttal · Reviewer_JoQZ · 2026-04-01
> >
> > Thank you for engaging with my comments.
> >
> > Re the Weaknesses, the paper does occasionally come across as dismissive of LLMs entirely, so a shift in the tone of the paper would help it align with the authors' views as expressed in this rebuttal.
> >
> > Thanks also for the examples to papers with good practice. As I raised in W2, and was also identified by Reviewer MANE, it is important to support the claim that many recent LLM+planning works fall prey to pitfalls with citations, rather than assuming it implicitly. This can be done with citations either to the works themselves (as outlined in the rebuttal to Reviewer MANE), or with citations to prior works that establish this. In my opinion, at least a compressed verison of this should appear in the introduction.
> >
> > Re. C1 and C2, I agree with the points made. I would recommend adding some of these nuances to the paper.
> >
> > I will be raising my score as my queries have been resolved and I support the acceptance of the paper.

---

### Official Review · Reviewer_Uced · 2026-03-12

**Significance:** 3
**Argument Clarity:** 3
**Rating:** 5
**Confidence:** 4

**Questions:**

Could the authors expand on the following:
- more assessments of tasks where the sequence of decision making matters and when an POMDP formulation of a problem may not satisfy this constrain?
- what is the difference between solution and valid plan, i.e., sound vs complete planners?
- it would be great if the authors could add more discussion to current work as done in Fig 1 to more actively discuss their position on what and where works are and are not consistent with planning nomenclature.

**Alternative Views Section:**

Yes

**Compliance With Llm Reviewing Policy A Conservative:**

Affirmed.

**Discussion Potential:**

3

**Final Justification:**

The rebuttal made convincing arguments and important clarifications, which address my concerns

**Paper Summary:**

This paper first summarizes the rigor in planning literature in terms of nomenclature, datasets and evaluations and contrasts it with the claims made by so called LLM planners. The central theme is that while several anecdotal claims are made about LLM's planning abilities based on certain benchmarks, the definition of planning is somewhat narrow, underspecified, often confounded by other skills required and therefore lacking in the evaluations and strengths of claims made.

**Position:**

Yes

**Position In Title:**

Yes

**Related Work:**

2

**Strengths And Weaknesses:**

Strengths:
- The authors make several thought provoking points about LLM evaluations with regards to their planning abilities. This paper can serve as both a tool for researchers designing benchmarks, reporting model evaluations, reviewers, or anyone making scientific claims about "planning abilities" and also as a good reference for literature in conventional planning that predates transformers and modern LLMs.
 - The authors some intriguing distinction for planning tasks (where the sequence of decision making matters) versus tasks that are solvable by planners such as SAT and sudoku.
- The authors also make important points about contamination during evaluation with regards to models memorizing PDDL and presenece of PDDL generators as a reasonable solution that can be easily adopted. Another such point is Pitfall 7.2 discussing the nature of planning tasks evaluated and that each solution is not of the same quality. Metrics such as time to solution can be particularly impactful with regards to current set of "thinking" models, which generate long text prior to the action prediction.

Weaknesses:
- Some of the nomenclature and discussion seems very specific to planning literature and could be made more accessible to aid in understanding of the paper. See clarifications in questions.

**Support:**

2

---

> ### Author Rebuttal · Authors · 2026-03-31
>
> We thank the reviewer for the constructive feedback.
>
> **Q1.** We hope that we interpreted your question correctly.
>
> On when the order in which decisions made matters:
> one might distinct problems where it does not matter how you get to the goal, only what the goal state is, or it might be important how you get to the goal, which actions were used, but these actions are independent, in the sense that it is not important in which order these actions are executed.
> Sudoku or crossword puzzles, or the nurse scheduling problem are prominent examples. In these cases, most modern classical planners, that perform an informed search in the problem state space will perform poorly, as they will explore the exponential number of orders in which the goal is achieved. The better performing solvers for such problems will be CP/MIP or SAT solvers, which perform a search in a different space. Thus, the choice of strong and more relevant baselines or basis for the solution depends on understanding the property of the problem being solved.
> The opposite is also true, if the problem has inherent sequential nature, CP/MIP/SAT solvers would not be a good fit, as a typical transformation into the input to these solvers would include a copy of the problem per possible sequential step, significantly increasing the input size.
>
> On POMDPs and representation size:
> POMDP is a rich mathematical model for partially observable probabilistic planning, which can encapsulate a lion share of planning problems, inccluding the examples above. One could explicitly describe a planning problem in a POMDP, but such an explicit representation can be exponentially larger than a compact representation in a planning language. One example when that can happen is independent information gathering actions. These actions can be invoked in any order and n such actions would be contributing a factor of O(n!) (times n!) to the POMDP size, while contributing only an addition of O(n) to a PDDL representation.
>
> **Q2.**
>
> > A planner that is guaranteed to only return a valid plan is a sound planner, and a planner that is guaranteed to return a solution if one exists is a complete planner.
>
> In this context, a plan and a solution are synonymous. A sound planner is one that, if it terminates, its returned object is guaranteed to be a valid plan. A complete planner is guaranteed to terminate and return a plan if one exists.
>
>
> **Q3.**
> We intentionally avoid paper-by-paper critique. Many works in the LLM-planning literature do not clearly state the planning problem, assumptions, or guarantees they target in standard planning terminology. Our goal is to encourage a more consistent way of situating such work.

---

> > ### Author Rebuttal · Reviewer_Uced · 2026-04-02
> >
> > I thanks the author for their responses, I believe most of these comments can be incorporated in the writing and will make the paper stronger. I will increase my score to support acceptance.

---

### Official Review · Reviewer_8Y5x · 2026-03-13

**Significance:** 4
**Argument Clarity:** 3
**Rating:** 5
**Confidence:** 4

**Questions:**

- What are the most important areas/pitfalls? What should the community address first?
- LLM-planners rely on iterative generation, tools, feedback from their environments: how should evaluation frameworks isolate the value of planning itself from the core capabilities of LLMs (e.g. language understanding, reasoning, etc.)
- How should the frameworks proposed be adapted for open-world use (natural language, partially observed)?A Classical planning assumptions don't hold here.

**Alternative Views Section:**

Yes

**Compliance With Llm Reviewing Policy A Conservative:**

Affirmed.

**Discussion Potential:**

3

**Final Justification:**

I maintain a positive view of the work and the rebuttal addressed any concerns I had.

**Paper Summary:**

The paper argues to bring the bring the 'rigorous' planning research and progress from the automated planning community into the modern LLM era. The paper provides a detailed overview of preliminaries for understanding planning, available tools, data, and how to evaluate planners, identifying benefits and pitfalls in each domain.

**Position:**

Yes

**Position In Title:**

Yes

**Related Work:**

4

**Strengths And Weaknesses:**

### Strengths
- The paper is well written and structured. The position is clear and explicit.
- Alternative views are identified and cover important objections, and the authors debate them well.
- The position has timely relevance to the broader ICML community and would promote productive discussion.
- Related work and an outline of the planning landscaping is well done/grounded and provides context for researchers to understand the position and engage in discourse. It serves the intended purpose of handbook for planning-based research.
- Calls to action and concrete recommendations are clear.

### Weakness
- Engagement with modern LLM-planners or agent-based frameworks is limited. It would have been beneficial to add more concrete connection between the two. For example, how these classical methods interact with modern world models, multi-agent systems, etc.
- The work does a good job of identifying various pitfalls and outlining the planning landscape, however this is a little exhaustive. Where priorities lie would help refine discussion.

**Support:**

3

---

> ### Author Rebuttal · Authors · 2026-03-31
>
> We thank the reviewer for the constructive feedback.
>
> **Q1.**
> We believe that much focus should be given to the design of the experimental evaluation, including comparison to strong relevant baselines, evaluating on a variety of domains, and testing on instance families that scale systematically in difficulty.
>
> **Q2.**
> Examples of relatively isolated planning benchmarks are classical PDDL benchmarks, especially IPC domains, where planning is defined formally, and correctness is independently verifiable. If natural-language interaction is desired, benchmarks derived from PDDL, such as AutoPlanBench or ACPBench, are better suited to isolating planning than open-world agent benchmarks, since they are grounded in an underlying symbolic planning task, although they still introduce some language-understanding overhead.
> Introducing new, previously unseen PDDL domains can strengthen the evaluation design, and AutoPlanBench supports an automated generation of natural language descriptions for PDDL domains.
> A vide collection of evaluated domains and an obfuscation of the domains can further strengthen the evaluation design.
>
> **Q3.**
> That is a very interesting question.
> Several planning-based benchmarks already provide the natural language representation. While we are not aware of such benchmarks that extend beyond the classical setting, in principle, nothing prevents non-determinism or a natural-language representation that provides only partial information. In fact, it should be easy enough to control the amount of information exposed to the agent, allowing for a fine-grained evaluation of the corresponding agents' abilities.

---

> > ### Author Rebuttal · Reviewer_8Y5x · 2026-04-03
> >
> > I retain my acceptance.

---

### Official Review · Reviewer_MANE · 2026-03-18

**Significance:** 2
**Argument Clarity:** 2
**Rating:** 2
**Confidence:** 3

**Questions:**

1. Can you provide a systematic overview of the relevant LLM-based planning literature and use it to argue that LLM-based planning is not rigorous in the classical planning sense?
2. Can you provide concrete examples showing that failure to incorporate insights, etc., from planning leads to LLM-based planning issues?
3. Can you substantiate how incorporating existing planning insights, etc., would address the issues in Question 2?

**Alternative Views Section:**

Yes

**Compliance With Llm Reviewing Policy A Conservative:**

Affirmed.

**Discussion Potential:**

2

**Final Justification:**

While the paper provides an informative overview of automated planning aimed at the ML community, the paper primarily provides a tutorial on the subject rather than arguing for a position. I maintain that the core position of the paper -- that LLM-based planning is not rigorous and needs to be made so -- is not adequately substantiated. Indeed, a central takeaway from the author rebuttal is that this position has already been established in prior work. For this reason, though the paper has value as a tutorial, its value as a position paper is weakened. I therefore keep my score.

**Paper Summary:**

The paper's position is that LLM-based planning research needs to be made scientifically rigorous, and that incorporating existing insights, methods, tools, and benchmarks from the well-established automated planning community will help establish rigor in LLM-based planning research. A detailed overview of automated planning terminology and related works is provided. A wide range of important planning methods and their pitfalls are described. An overview of classic planning benchmark tasks is given and available planning tools are provided. Drawbacks of LLM-based planning approaches are mentioned.

**Position:**

Yes

**Position In Title:**

Yes

**Related Work:**

1

**Strengths And Weaknesses:**

**Strengths**

The paper provides a detailed, highly informative tutorial overview of automated planning methods, tools, and benchmarks. This overview of planning would be a very useful resource for members of the community interested in better understanding and applying planning work in their own research.

**Weaknesses**

The position of the paper relies on some core underlying assumptions, including:
1. LLM-based planning research is not rigorous in the sense of being firmly grounded in existing, rigorous planning work;
2. lack of rigorous grounding in planning leads to serious drawbacks of LLM-based planning methods in practice;
3. incorporating rigorous grounding in planning will improve LLM-based planning methods.

These statements are implicitly assumed in the paper, instead of being explicitly substantiated and argued for. Though these claims may be more or less accurate, since they are core parts of the overall position of the paper, they must be clearly substantiated in order for the argument for the position to be convincing. Specifically, while a systematic overview of the automated planning literature is provided, no systematic overview of the related LLM-based planning literature and its drawbacks is provided, undermining 1. In addition, no examples and relevant citations in support of 2 are given, and a clear research roadmap describing how 3 might be achieved is not provided. A systematic overview of LLM-based planning methods is needed to both (a) properly situate the paper's claims in their broader context, and (b) sufficiently address existing works that have also considered the relationship between formal planning methods and LLM-based planning (e.g., the 2024 ICML position paper "LLMs Can’t Plan, But Can Help Planning in LLM-Modulo Frameworks" by Kambhampati et al.).

**Support:**

1

---

> ### Author Rebuttal · Authors · 2026-03-31
>
> We thank the reviewer for the constructive feedback.
>
> **On weaknesses**:
>
> The assumptions made are slightly different:
>
> 1. A nontrivial subset of recent LLM-planning work repeats methodological pitfalls already known in automated planning, due to similar mistakes made in prior work in that community.
> 2. These pitfalls primarily threaten scientific rigor and make results harder to compare and interpret.
> 3. The automated planning community has developed methodologies, tools, and established evaluation practices that can help alleviate these issues.
> 4. More broadly, established planning formalisms, tools, and evaluation practices, many of which have already been developed and are readily available in the automated planning community, can help make such work more rigorous and reproducible.
>
> The first point was already established in prior work (e.g., [1]), and we are happy to see that the reviewer does not express any doubts about the correctness of the other points.
>
> A variety of systematic overviews of hundreds of LLM planning methods already exist (e.g., [2,3,4,5,6]), allowing us to focus on the position we are making in this work.
>
>
> **Questions:**
>
> **Q1.** As we mention above, this was already established in prior work (e.g., [1]).
>
> **Q2,3.**
> While we mention the many common pitfalls known to the planning community, we avoided giving concrete examples from LLM planning literature, albeit we did cite some of the papers. Our choice not to enumerate specific faults in the LLM-planning papers that fall into these pitfalls was intentional. This paper was meant as a constructive, forward-looking position paper rather than a critique of particular authors or methods.
>
> Arguably, these pitfalls recur often enough to matter at the level of community norms and methodology. We therefore chose to focus on general patterns and lessons from automated planning that can help improve future work, while keeping the paper's tone collegial and forward-looking.
>
> While we can be more explicit in our response and present concrete examples, we would prefer not to put them into this paper.
> One prominent example is the now-famous Tree of Thought paper, Yao et al, NeurIPS 2023 (6k+ citations), which we refer to in a few pitfalls.
>
> 1. The method is an LLM-guided search procedure, but the paper does not discuss soundness or completeness, nor does it make explicit that the method may return invalid outputs or fail to find a solution even when one exists, which is directly related to Common Pitfalls 4.1, 7.1, and 7.3.
>
> 2. The work also does not provide a planning-style complexity analysis, which again relates to Common Pitfall 7.1, and later analysis shows that ToT-style methods can be extremely expensive in LM-call complexity.
>
> 3. In the released Game of 24 code, the validation logic is embedded in the task code rather than separated as an independent validator, and it does not fully enforce the intended action language, since it checks digit usage and whether the expression simplifies to 24 but does not explicitly restrict the allowed operators to {+, -, /, *}; this connects to Common Pitfalls 5.3 and 7.3.
>
> 4. The evaluation is centered on simpler LLM prompting baselines rather than established planning systems from the planning literature, which is closely related to Common Pitfalls 4.2 and 7.1.
>
> 5. Finally, the benchmark itself is based on scraped public instances, which raises the contamination and memorization concerns discussed in Common Pitfall 5.2, and later work has shown that such public static benchmarks can give an overly optimistic picture of LLM planning performance (performance can drop from around 80% to around 20% when moving to generated instances instead of internet scraped ones).
>
>
> Each of these pitfalls, if addressed, would substantially improve that work.
>
>
>
> [1] Katz et al, NeurIPS 2024, Thought of Search: Planning with language models through the lens of efficiency
> [2] Wei et al, ACL 2025, PlanGenLLMs: A Modern Survey of LLM Planning Capabilities
> [3] Huang et al, Arxiv 2024, Understanding the planning of LLM agents: A survey
> [4] Pallagani et al, ICAPS 2024, On the Prospects of Incorporating Large Language Models (LLMs) in Automated Planning and Scheduling (APS)
> [5] Aghzal et al, Arxiv 2025, A Survey on Large Language Models for Automated Planning
> [6] Cao et al, Arxiv 2025, Large Language Models for Planning: A Comprehensive and Systematic Survey

---

> > ### Author Rebuttal · Reviewer_MANE · 2026-04-03
> >
> > Thanks to the authors for their response. The concerns outlined in my review remain largely unaddressed, however, and I maintain that the core position of the paper is not adequately substantiated. The paper reads more as a tutorial on automated planning aimed at the ML community than as a position paper carefully arguing for its position. I keep my score.

---

### Decision · Program_Chairs · 2026-04-30

**Decision:**

Accept (regular)

**Comment:**

The overall review is strongly positive, with three reviewers supporting acceptance and indicating that their concerns were resolved after rebuttal. One reviewer maintained a negative score. I agree that the paper is somewhat heavily toward tutorial and synthesis, and that the argument would be stronger with more explicit substantiation of pitfalls through concrete LLM-planning examples in the main text. However, the paper still makes a timely and useful position paper contribution by proposing a clear call for methodological rigor, grounding that call in longstanding planning practice, and identifying concrete pitfalls and evaluation norms that can help the broader research community.